# Hazard Reduction in Deep Excavations Execution

Mateusz Frydrych [ID], Grzegorz Kacprzak [ID] and Paweł Nowak *[ID]

Civil Engineering Faculty, Warsaw University of Technology, 00-637 Warsaw, Poland;
mateusz.frydrych.dokt@pw.edu.pl (M.F.); g.kacprzak@il.pw.edu.pl (G.K.)
* Correspondence: p.nowak@il.pw.edu.pl; Tel.: +48-22-2346515

**Abstract:** In this article, the authors consider a completely new approach in design, which is related to the use of previously un-adapted technologies known to bridge engineering in geotechnical issues for prestressing of diaphragm wall during deep excavations execution. The bridge technology described here is the prestressing of concrete structures. Hazards related to deep excavations and methods of digging them, such as the ceiling method and top&down method, are presented. Current problems in supporting deep excavation slopes are related to the use of extensive quantities of materials (such as steel struts, ground anchors, or concrete and reinforcement steel). The authors' method helps to achieve a higher level of sustainability, which is important in a modern approach to geotechnical engineering. The non-linear arrangements of the cables according to the occurrence of the prestressing moments for a given phase are presented. Results related to numerical analysis—showing the correctness of the method and cost optimization results, showing possible savings are presented. The article is a part of the set. In the second (already published) article titled "Modern Methods of Diaphragm Walls Design", the authors present the concept of the calculation methodology for diaphragm wall design.

**Keywords:** hazards; sustainability; deep excavations; diaphragm wall; prestressing





## 1. Introduction

The design and construction of vertical structural elements, such as retaining walls, is an integral part of the design process in the case of buildings with underground floors. It is common that the retaining elements, e.g., in the case when diaphragm walls are a necessary part of the slab-pile foundation, the important process of their design and implementation on a construction site should not be overlooked. The design process is based on laws, regulations, and design standards. It is impossible to meet the basic design requirements without taking into account the cooperation between the considered structure and the ground as it is stated in Marschalko M. et al. [1] Obviously, there are various types of structures that are placed on the seabed, bedrock, or in unusual soil conditions. In each of these cases, the design process must consider the need for transferring the loads from the structure to the ground medium [2]. It is not possible, in the investment and construction process, to not consider contact between the structure and the ground, because taking into account the aspects related to the foundation of any structure, it is highly probable that problems will occur during the construction and maintenance of the object. This is due to the need to adopt design assumptions related to the substrate for a given investment. Soil conditions are often very unpredictable, and the correct interpretation of the subsoil parameters is one of the most difficult steps in designing a foundation. The report from the Building Research Institute (Poland-ITB) on building disasters and failures from 2014 showed that 41% of geotechnical failures were caused by improper design [3,4]. It is worth noting here that the entire responsibility for the design documentation of the structure of the considered object is carried by the designer of the structure, including for the adopted solutions based on the geotechnical parameters that are included in geological studies. It is the designer who ultimately decides whether the soil parameters proposed

by the geologists are appropriate to be used in the calculations. This direct correlation between the work related to the ground recognition and the preparation of the project is so important that it becomes natural to take an approach that does not encourage savings on geotechnical research.

In geotechnical design environments, it is believed that a running meter of drilling will always be cheaper than making an additional meter of pile foundation, and that additional verification of the selected design solution made by an external consultant will always be cheaper than taking corrective action in case of geotechnical failure. Unfortunately, the reality of the investment and construction process is different. A very wide range of ground recognition options is known, but investors trying to generate savings on the planned investment are reluctant to agree to the guidelines and recommendations of designers regarding the need for performing a network of geological surveys, soundings, and analyses. For the reasons listed above, it is extremely important to find the right method in order to minimize geotechnical risk. This method should be inexpensive to implement and, at the same time, must produce the best possible measurement results. In this paper, the authors present popular methods of constructing underground stories, taking into account the geotechnical risk. They also show known ways to minimize the risk in structural solutions, related to the construction of ground anchors and the use of steel struts [5]. As a non-structural method, the authors describe the use of geodetic and measurement monitoring [6]. Groundwater aspects were adopted into the model as additional external loads on the stabilized water level. The diaphragm walls themselves have waterproofing capabilities, so water does not enter the excavation through the wall. The only place where water can enter the excavation is at the bottom of the excavation [7]. A generally accepted solution is to pump water out of the excavation using depression wells or well point systems. Additionally, in the event of significant rainfall, surface pumps are used so that excavation work can be continued. In the analyzed model, water could be considered as a wall pressure for the calculations, but to simplify the presentation of the considered concept, authors show results without water pressure. That pressure could be part of the load combination, but it can be excluded in the case of cohesive soil analysis due to the consolidation phenomenon when pressure is strictly connected with groundwater conditions The results shown in the following sections of this paper illustrate a situation where the groundwater table is lowered by using a depression well, so the water is not a load with a continuously operating drainage system that could be disassembled after the building basement is completed. It is not necessary to include water evaporation in the model (but it is possible in further research) because prestressing is a temporary element in the diaphragm wall design [8]. Prestressing the wall in the temporary phase allows other strutting methods, such as ground anchors or struts, to be dispensed with and is not needed once the underground part of the building is constructed [9]. Section 1 shows a general overview of geotechnical issues of deep excavation, including groundwater influence. Section 2 shows popular methods of deep excavations execution: celling and top&down methods as a base of comparison with suggested the prestressing of diaphragm walls method. In Section 3, the authors describe popular methods of monitoring geotechnical structures on construction sites as tools for hazard limitation. Section 4 presents the new innovative concept of the prestressing diaphragm wall method with validation in FEM software (Plaxis, Sofistik), as well as the results and technological aspects of this solution. A comparison shows the correctness of new methods' assumptions. In Section 5, the authors discuss cost savings with the use of the proposed solution. Section 6 contains the summary.

## 2. Theoretical Basics of Research

Research shows that the typical budget for a ground identification still does not exceed 0.02% of the investment costs in Poland [10], which puts the Polish construction industry in one of the last places in Europe. Spending such a limited amount of money on preconstruction geotechnical investigation is very hazardous.

Over the years, especially after World War II, prestressing technology has gained popularity, mainly because of advancements in the development of construction materials. Originally, prestressing technology was used for bridge engineering because its use directly translated into an increase in the span of the structures built at that time. Today, prestressed concrete is widely used in the design of bridge structures, mainly thanks to the possibility of achieving significant spans while using less steel and concrete.

Prestressed concrete has both advantages and disadvantages, and it is not justifiable to use this design method in all cases [11,12]. Over the years, the idea of prestressed concrete has been very developed, and it is widely used not only in bridge construction but in other fields as well [13,14]. There is now a growing trend towards the use of prefabricated prestressed elements, which translates into a number of advantages. The use of prestressed concrete in slab and wall elements is becoming very popular. Just as in the case of bridges, it is possible to obtain a significant span with the economic use of less concrete and steel and with a reduced construction height, which is very advantageous in office and residential investments [13,15].

The concept of prestressing is also used in geotechnical considerations, e.g., in the design and manufacturing of ground anchors. There is no issue with obtaining a larger span or using less steel because it is a completely different type of construction. On the other hand, the idea of the work of this element is similar. It "artificially" introduces additional forces to the considered element, which are meant to counteract the naturally occurring loads. The main issue considered by the authors of this paper is the combination of the known method of designing prestressed bridges with the methodology of designing and constructing diaphragm walls [16]. The purpose of this operation is to generate technologically more favorable results that will translate directly into a lower cost for the investment. Embedding a prestressing cable into the diaphragm wall can significantly reduce the amount of reinforcing steel needed, which directly translates into lowering the cost of making the diaphragm wall. Another much more important advantage of using this type of solution is technological optimization [17]. All these processes improve the course of construction and allow the investment to be carried out faster and cheaper than in the case of using the classic approach for designing the foundation. Resignation from the need to use additional bracing, wall anchoring, or making an expensive expansion ceiling is the sustainable solution. Everything depends on the imposed investment conditions because the use of this innovative method will not be justified in every case. This paper aims to show the effects of adapting the bridge method in the design of diaphragm walls as an innovative methodology for designing a foundation. The authors will focus on point investments because those are the types of building objects that the described solutions are dedicated to. As an example of the application of this design method, a common variant of the implementation of the foundation in a dense urban development for high-rise buildings was selected [18]. It comes down to the construction of four or more underground stories in the sheathing of diaphragm walls, using the ceiling or top&down method.

The most common methods of constructing underground spaces in buildings are as follows:

1.  Ceiling method (ceiling)—the method was first used in the 1960s in the construction of the Milan metro. In the literature and engineering practice the name "The Milan method" is also used. The undoubted advantage of this method is the possibility of using it in the city centers with dense buildings around the investment. Due to the very favorable price–execution ratio, and at the same time the large availability on the market of the needed equipment, it is currently the most popular method of constructing underground stories in dense urban development. Of course, for the sake of specific implementations, modifications are used (classic method, semi-ceiling method) in the form of a special arrangement and form of expansion ceilings (e.g., ring arrangement with one large technological hole in the middle of the ceiling—semi-ceiling method), anchoring parts of the housing, or using hybrid solutions, in which part of the wall is supported by a strutting ceiling, some with steel struts, or part of

the casing is anchored in the ground with the use of ground anchors. There are many and it all depends on the number of underground floors and technological conditions. Undoubtedly, the greatest design and implementation challenge is developing the concept of investment implementation, in which the adopted solutions allow performing the necessary construction works as quickly as possible at the lowest possible costs. The main idea of the ceiling execution method comes down to:

- Execution of the excavation casing from diaphragm walls;
- Deepening the excavation to the required level of strut or wall anchoring or to the level at which the expansion ceiling is meant to be created;
- Depending on the depth of foundation and the conditions of a specific investment, it is possible to construct temporary columns that will support the ceilings until the columns/target support are already at the stage of the first anchoring. The columns usually have their support in the bars, which are made from the level of the first strut or the working platform. It depends on the conditions of implementation. (Figure 1a,b).
- Then the formwork is laid and a strut ceiling with a technological opening is made. When the concrete mix is hardened, the process of removing it from under the previously made expansion ceiling begins. The area under the ceiling is leveled to the next level of the floor below. The next steps are analogous, i.e., making the next expansion ceiling at the ordinate assumed by the designer and removing the soil through the technological hole. These activities are repeated depending on the assumed number of strutting ceilings until the target foundation level is reached, where the foundation slab is made (see Figure 2).
- After the foundation slab is constructed, all vertical elements, such as columns and core walls, are made from the foundation slab level to the 0 level. During this process, the formwork is traditionally made, and the technological holes are concreted. If there are struts in the technological openings (what can happen if there are large dimensions of the basement), they are dismantled. Work continues until the "closure" of the underground on level "0" (see Figure 3).
- After making the necessary vertical elements and concreting the technological openings, it is possible to remove the temporary columns. For this purpose, the column itself and the heads on which the ceilings rested are cut off at the level of the foundation slab. Depending on the number of underground floors and the capacity of the crane, it is a common practice to cut the columns into smaller sections.

2. Top&Down Method: Another method of creating deep excavations is the top&down method (see Figure 4). The biggest advantage of this method is the implementation time, as it is significantly shorter than the ceiling method. This is the main feature that distinguishes this method, i.e., the simultaneous construction of both underground and overground stories. After the diaphragm walls are made, temporary columns are made, similarly to the previously described ceiling method. The level of execution of these poles depends on the conditions of a given investment. These poles are based on bars or, less frequently, on piles. The next step is to make the main floor slab of the facility. According to the adopted solution, it is decided whether it is a level "0" or lower. In this slab, just like in the ceiling method, a technological hole is left, which allows for extracting spoil from under the ceilings of underground floors. The biggest difference which characterizes this method is the fact that while extracting the spoil from under the underground stories (the work is carried out in the same way as in the case of the overhead method), vertical floors are made using common methods. The entire workload is carried by temporary columns, up to the construction of vertical structural elements from the level of the foundation slab to the level of the main floor slab described above. This method is much riskier than the ceiling method because it is very sensitive to changes in the schedule and performing all activities must be very precisely defined and supervised.

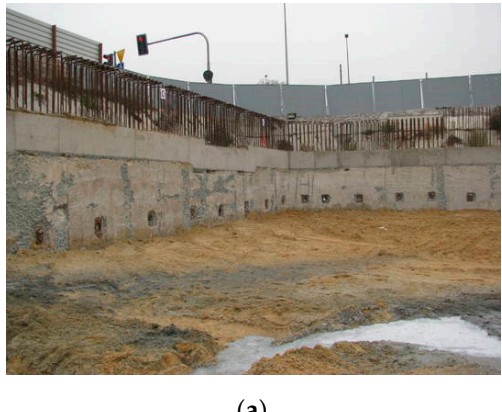 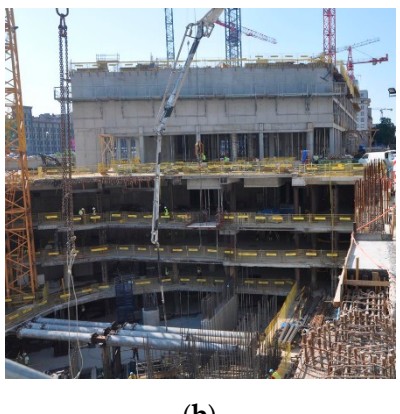

(**a**) (**b**)

**Figure 1.** (**a**) Digging the trench to the level of the first strut. (**b**) View of the temporary columns placed in the ground on which the target ceilings rest, including expansion ceilings. Adapted from ref. [19,20].

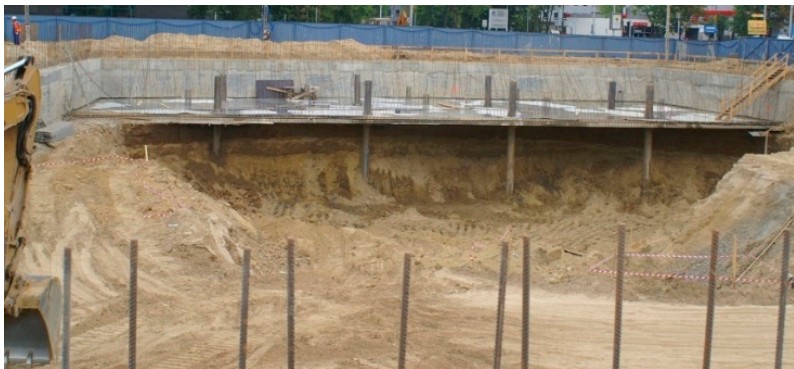

**Figure 2.** The process of removing the spoil from under the expansion ceiling was made. Temporary columns supporting expansion ceilings are visible. Adapted from ref. [21].

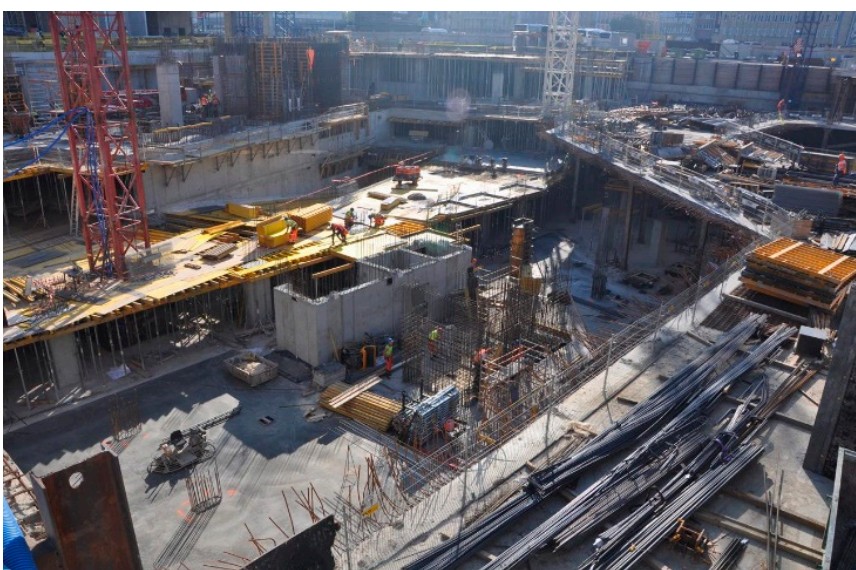

**Figure 3.** The process of making vertical elements after the foundation slab was made. At the same time, traditional horizontal formwork is made, and the technological opening is concreted. Adapted from ref. [20].

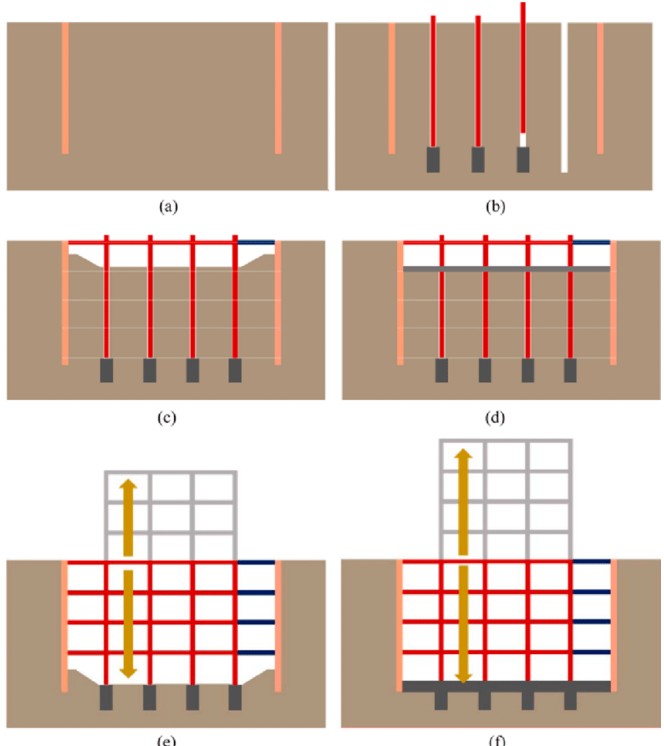

**Figure 4.** The technology of investment implementation using the top&down method: (**a**) making diaphragm walls; (**b**) construction of temporary columns; (**c**) execution of the main "0" slab; (**d**) removal of spoil from under the ceiling, constructing further underground ceilings; (**e**,**f**) simultaneous execution of the target over-ground structure. Adapted from ref. [22].

## 3. Hazards Occurring during Deep Excavation Execution

The underground construction of buildings carries a high risk, which should be minimized at the design stage and controlled during construction. Risk is related to various factors because every construction site is different, but all construction sites have one thing in common—geotechnical aspects. It means that all buildings have foundations and because of that, excavations need to be executed. Deeper excavation means that the risk related to its execution is higher.

Risk minimization, while following the principles of sustainable development, can be done in several different ways. The main division of risk minimization boils down to two categories: direct and indirect collateral [23]. Direct risk minimization includes, among others, methods related to structure monitoring. Monitoring allows directly checking deformations and stresses in the structure in real-time using built-in sensors. There are many different sensors that are precisely described [24]. The variety of sensors offered by the market is huge, so it is possible to measure any design for different verification conditions. It is particularly important for the trench support, as the safety of the construction works depends on its stability. If the excavation support is monitored, it is possible to react quickly in the form of adding an additional strut or earth buttress. The example [25] clearly shows the possibilities of monitoring individual structural elements. The type of sensors and their arrangement scheme depends on the type of housing. The lining considered by the authors—a diaphragm wall—is an example of a reinforced concrete lining, which allows for the execution of a deep excavation suitable even for several underground floors [26]. It is particularly important to monitor this type of support because it is a complex system, which is subject to many different loads that change in time, such as loads caused by construction equipment at the ground level, operational load inside the trench, soil pressure, thermal loads, and groundwater pressure. It is extremely important to integrate the sensors correctly and calibrate them. Any sensor that is properly connected can show the result. However, it

is not true that every result is correct; therefore, in addition to monitoring with sensors, it is important to monitor the housing by means of geodetic measurements [27].

Geodetic measurements are a special example of risk reduction and influencing the sustainable way of conducting investments. There are different types of surveying methods that can be used to monitor structures during construction. The use of sensors embedded in the structure is very popular. These are small elements that collect deflection data and through them, it is possible to assess the displacements and stresses of the structure. The measurement session carried out at specified times is designed to verify the readings from the sensors placed in the construction. The results of the surveying session express the displacements of the structure. Once the displacements and material data, in the form of deformation modules, are known, it is possible to determine the stresses. The determination of stresses allows verifying the behavior of the structure; thus, it is possible to check on an ongoing basis that the allowable stresses are not exceeded, which may lead to a construction disaster [22]. Another well-known method is the installation of inclinometers into diaphragm walls. These are vertical measuring elements mounted in a sheath that allows for monitoring of horizontal displacements along the length of the entire inclinometer. It is possible thanks to the installation of reference benchmarks to measure displacement, control clinometry, and check the height and level of the relevant diaphragm with the use of leveling measurements [28].

To minimize the risk and at the same time achieve sustainability goals, the use of indirect methods is a very good choice. Indirect methods are those that boil down to reinforcing the existing solutions or to considering whether the applied conceptual solution is the best. Execution of each investment is thought out in advance. This means that it is necessary to analyze every possible technological variant in the form of securing the excavation, maintaining safety, calculating the number of expanding or anchoring elements for possibly, and cheap but, at the same time, safe construction of the underground part of the building [29].

There are many options for securing the stability of diaphragm walls. The most popular is using steel in the form of steel struts or making ground anchors. Another very popular method described in Chapter 2 is making expansion ceilings. Each of these technologies has positive and negative aspects to their use. In the case of steel struts, it should be known that their installation is particularly dangerous due to the size and weight of the elements. In addition, the installation of struts is possible only with the use of a construction crane, which is very expensive. Each strut assembly takes time, so construction time is longer, especially when the time for strut disassembly is considered.

This time when the crane, powered by electricity, is working means that it contributes to an increase in carbon dioxide in the air. Another serious disadvantage of steel struts is the fact that when using them as struts, one must be careful with reinforced concrete work in the basement of the building. Always remember that people are working under the struts and hitting the steel strut with the end of the crane may damage the strut and cause its fall.

As in the case of steel struts, ground anchors are time-consuming to manufacture, and their implementation requires using a special machine that is expensive to operate and contributes to increasing the amount of carbon dioxide in the air. The ground anchors must be stressed to achieve their load-bearing capacity, which means an external force must be applied to the tendons in the anchor. This is a dangerous process as a defective tendon can be hit at any time and the metal ring at the end of the anchor can be damaged. Due to the very high internal forces in the anchor, said element may cause harm to the workers installing the soil anchors. It is certainly not a solution that fits in with the idea of sustainable development.

One of the last methods of securing diaphragm walls against damage is the execution of expansion ceilings between the walls. This means making expansion ceilings on the ground while maintaining a special technological hole. This process is used to carry out further earthworks, as described in paragraph 2 for the roof method. This solution is very

unbalanced because the construction of an expansion floor faces two problems: optimal time usage and minimizing the amount of steel used. It takes a long time to build a strut ceiling because a large amount of reinforcement must be placed in it. Reinforcing steel is expensive and energy-consuming to produce, so any action that reduces steel usage contributes to sustainability. What is more, the resources that would be used in the construction of the expansion ceiling could be used for further construction elements, i.e., the investment would be faster. The big disadvantage of spreader ceilings is their geometry and the limitation of the possibility of working on the next underground floor. Their presence makes the work very dangerous, as workers and machines have to work under the ceilings. Additionally, expansion ceilings require additional supports, which usually are temporary columns. These are the steel elements that hold the expansion ceilings before the final reinforced concrete columns are erected (Figure 4). After the final structure is completed, it must be removed from the building, which takes a long time and consumes a lot of energy. Therefore, it is worth asking whether there is another way to secure the excavation using diaphragm walls to reduce the risk and increase the environmental friendliness of the solution.

## 4. New Concept of Excavation Slope Protection

Another innovative way to safely construct the underground floors while maintaining a sustainable approach is the use of prestressing in the diaphragm wall structure. This innovative concept makes it possible to dispense with ground anchors or steel struts. This is a huge technological advantage because, depending on the complexity of the investment, using steel struts can significantly hinder the implementation of, e.g., reinforcement works. It all depends on a given conceptual situation, but the lack of spacers is undoubtedly an advantage. It is worth emphasizing that the laying of the prestressing cable itself can have different effects. This feature can be used, depending on the specific needs, but in this case, the simplest variant was used, i.e., the prestressing cable was laid along the vertical axis of the diaphragm wall. The non-linear arrangement of the cable was according to the occurrence of the prestressing moments for a given phase, and in general, for the envelope, translates into a reduction in horizontal displacements and stresses in the reinforced concrete element. A natural consequence is reducing the quantity of steel needed to transfer the tensile stress in the element. Linear laying of the cable or placing it closer to the trench may have a positive effect on the stress system, which is responsible for the fulfillment of the scratch demands in the further design process [30]. In some concepts, it is the failure to meet the condition of not exceeding the crack value that significantly increases the amount of prestressing steel. The idea of this new system is presented in Figure 5a,b. This type of approach fits in with the idea of minimizing risk for a number of reasons.

First of all, it is possible not to install steel struts or to make ground anchors. Thanks to this, there is more space in the excavation, which allows saving time for subsequent construction works and it is possible to partially omit the making of the expansion ceilings. Reducing the time needed for the execution of the individual elements mentioned above brings with it consequences in the form of saving money. The construction of underground stories makes a lot of sense without struts, anchors, and expansion ceilings. Another advantage is that there is no need to use heavy construction equipment, such as excavators, cranes, or a ground anchor machine. Thanks to this, it is possible to reduce the carbon dioxide emitted into the atmosphere; thus, it is in line with the idea of sustainable development. The installation of prestressing in the diaphragm wall is an additional cost, but from the perspective of reducing other expenses, such as struts, ground anchors, or reinforcement of strut ceilings, it is an insignificant cost.

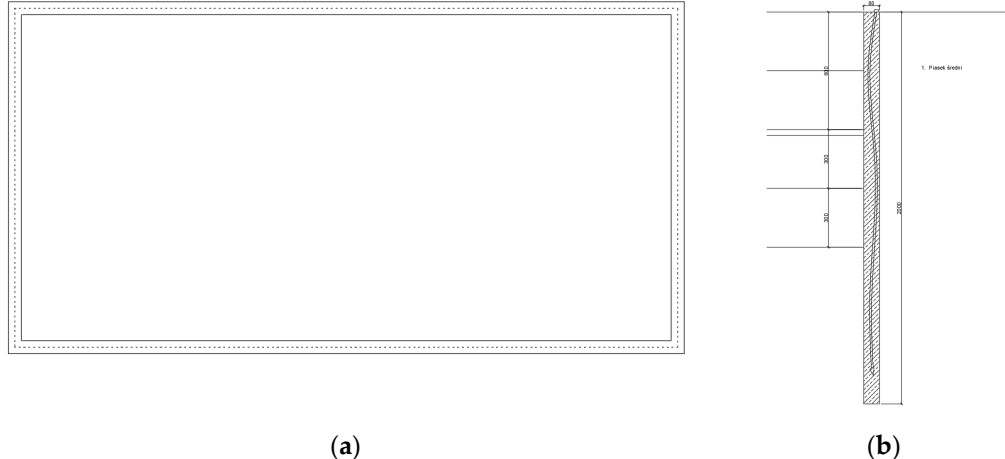

|  (a)  |  (b)  |

**Figure 5.** (**a**) Concept II—projection of the suggested solution. The excavation space is free from obstacles in the form of struts. The neighboring land is not occupied by ground anchors. (**b**) Concept II—a cross-section of the proposed solution. In this solution non-linear routing of the tensioning cable is possible. [own source].

The description of the used numerical model assumptions, such as boundary conditions, constitutive equations, the solving method, the material properties (including soil [31] and structural elements) are given in detail in the article "Modern Methods of Diaphragm Walls Design" *Sustainability*. 2021. Vol. 13, no. 24, p. 1–14. DOI 10.3390/su132414004. (presented by the same authors). The authors made a comparative model of Sofistik with Plaxis (a trusted digital environment used in everyday geotechnical analysis) in order to verify the modeling results. Both models were created to check the comprehensiveness of the suggested method and to assure the correctness of the results. Table 1 and Figures 6–9 show the same outcomes of both models (from Plaxis and Sofistik), so a few characteristic points as to the coordinates were selected and the exact results were compared at specific nodes, using the results generated by both environments in tabular form. The result of this comparison is presented in Figure 6.

**Table 1.** Comparison of the results of Plaxis–Sofistik displacements for the 3rd phase-4.0 m excavation. [own source].

| Node No. | Number According to the Figure 6 | Coordinates | | Displacements in Plaxis | | Displacements in SOFISTIK | |
|---|---|---|---|---|---|---|---|
|  |  | X [m] | Y [m] | u_x [mm] | u_y [m] | X [mm] | Y [mm] |
| 4627 | 1 | 4.000 | 0.000 | −1.920 | 0.001 | −1.567 | 0.820 |
| 3311 | 2 | 7.522 | 0.000 | −0.496 | 0.002 | −0.668 | 1.641 |
| 4487 | 3 | 4.000 | −2.600 | −0.417 | 0.003 | −0.364 | 3.030 |
| 2796 | 4 | 8.195 | −4.462 | −0.309 | 0.002 | −0.438 | 1.682 |
| 4478 | 5 | 1.423 | −4.000 | 0.231 | 0.008 | 0.159 | 8.070 |
| 4654 | 6 | 3.267 | −4.000 | 0.196 | 0.008 | 0.219 | 7.260 |
| 4542 | 7 | 4.000 | −4.000 | −1.114 | 0.005 | −1.358 | 4.030 |
| 2933 | 8 | 4.000 | −10.000 | −0.486 | 0.003 | −0.658 | 3.020 |
| 746 | 9 | 9.286 | −11.585 | −0.555 | 0.001 | −0.758 | 0.848 |

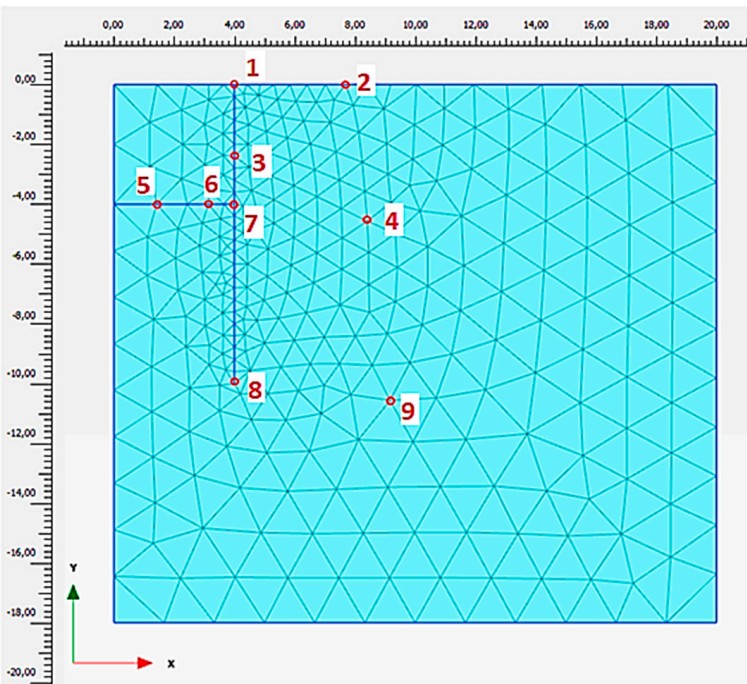

**Figure 6.** Model comparison-Plaxis-marking selected nodes for accurate comparison of results. Grey background color and blue color which presents the results are common for Plaxis postprocessor. [own source].

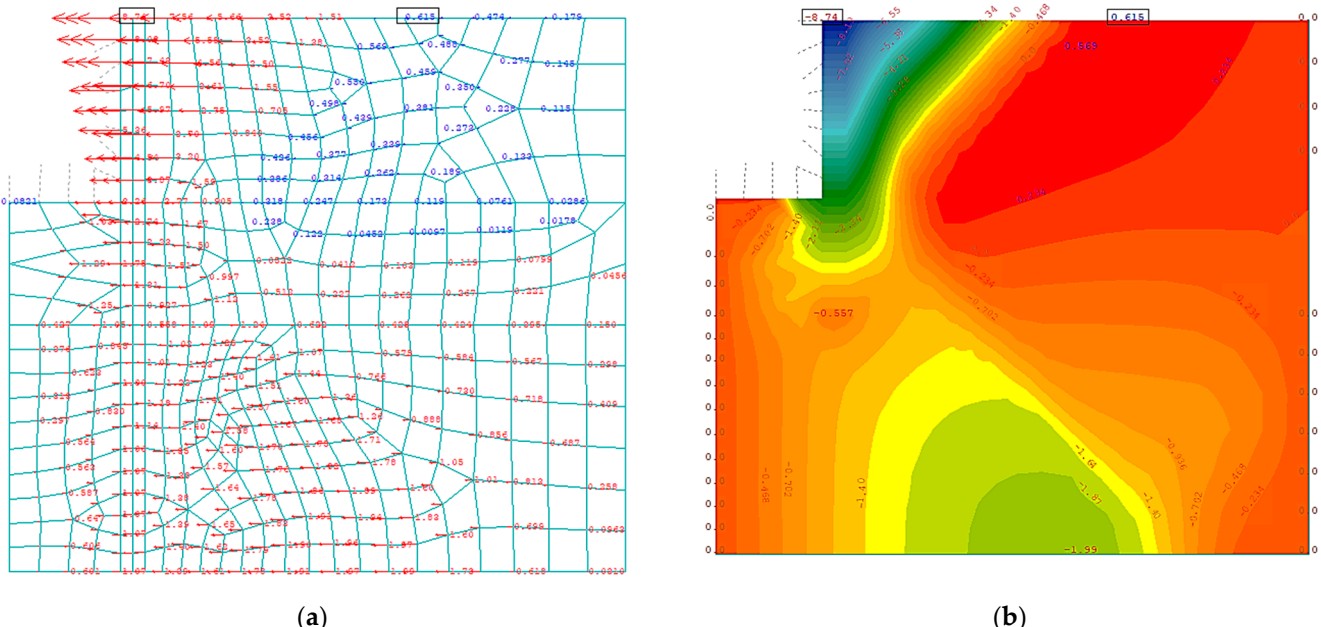

(**a**)                                                                                       (**b**)

**Figure 7.** (**a**) Sofistik-phase IV-excavation 6.0 m-horizontal displacements-vector form-max u_x for the top of the wall it is 8.7 mm-no prestressing. (**b**) Sofistik-phase IV-excavation 6.0 m-map of horizontal displacements-max u_x for the top of the wall is 8.7 mm-no prestressing. (Own source).

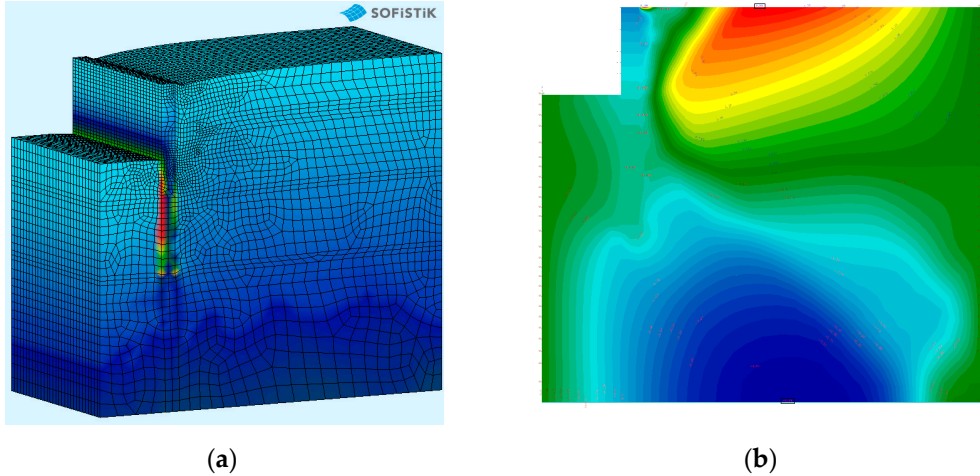

(**a**)  (**b**)

**Figure 8.** (**a**) Sofistik_visualization—phase IV—excavation 6.0 m—no prestressing. (**b**) Sofistik—phase IV—excavation 6.0 m—horizontal displacements—max u_x for the surface of the wall 3.06 mm—prestressing. [own source].

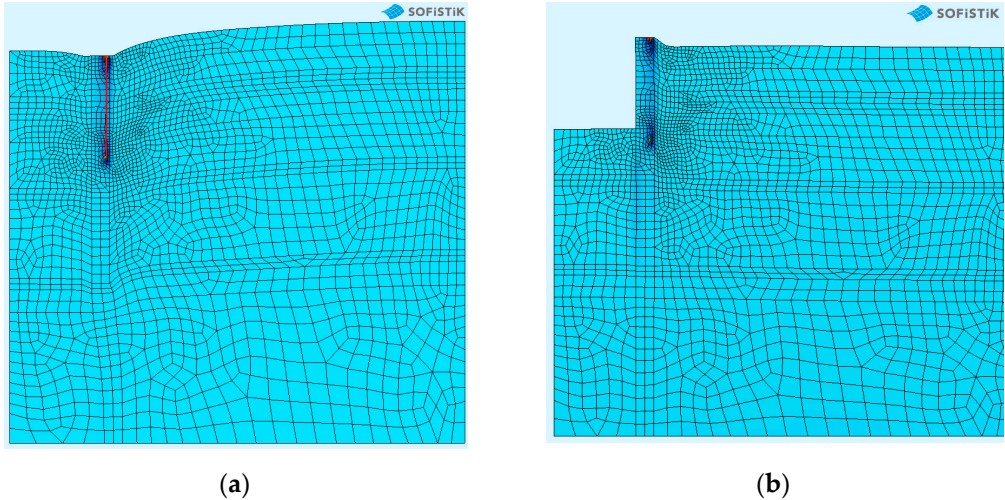

(**a**)  (**b**)

**Figure 9.** (**a**) Sofistik_ visualization—phase III—prestresing wall. (**b**) Sofistik_ visualization—phase IV—excavation 6.0 m with prestressing. [own source].

Validation shows the correctness of the method's general assumptions. The results of the adaptation of prestressing methods known in bridge engineering to the diaphragm wall are shown at Figures 7–9. After validating the model with regard to the soil environment and cooperation of this environment with the structure, and after checking the operation of the algorithm for prestressing reinforced concrete elements, it is possible to combine these issues into one complete model. Thanks to this, it will be possible to analyze a simple case of a prestressed reinforced concrete element loaded with earth pressure in one coherent environment (see Figure 7). Detailed calculations of the losses in prestress diaphragm walls were not taken under consideration as prestressing could be treated as a temporary phase, but it is the subject of further research.

The analysis above shows that, thanks to the use of prestressing, it was possible to significantly decrease displacements in the considered reinforced concrete element. The stresses were also decreased after introducing the prestressing force. These results prove that the prestressing of an element, which is a diaphragm wall, advantageously changes the internal forces and displacements of the element in terms of the dimensioning of the wall itself. The additional phase of prestressing the element directly affects the scratching, lowering it, which translates into a reduction in the general quantity of reinforcement

needed, thus minimizing the costs of building the wall itself. Another huge advantage is the possibility of making a deeper excavation, which is a huge advantage from a technological point of view. Conclusions from the application of this methodology and an example of the financial benefits of its use are presented in Chapter 5.

## 5. Costs Optimization in Deep Excavations Execution

In this chapter, the authors present the effect of applying prestressing in vertical structures of reinforced concrete diaphragm walls in terms of the optimization of cost management and technology of the work. Issues related to the valuation of construction works are often the source of the greatest risk (i.e., financial) from the perspective of the project implementation. Each design solution, each idea, and vision of a designer entails a series of activities that must be performed in order to implement this vision. To be able to determine the results of the methodology presented, a typical case was selected, a high-rise building, with three underground stories. The case described below is very general and shows the rules and the advantages of the discussed innovative design methodology in a simple way. Each costing process in geotechnical works should cover the soil conditions, the number of underground stories, the place of investment implementation, and the equipment lease. There are many financial variables, and it is not essential to describe all the relationships between these parameters in this article. The authors chose a general case that illustrates the idea of applying prestressing in vertical diaphragm wall structures and what advantages it brings. The general cost structure is similar in each discussed case, and the differences are underlined. The examples of excavation slope protection which are hazardous for work performance [32] are described in this chapter as:

- Concept I—traditional construction of the underground with the use of Concept Ia—ground anchors and Ib—steel struts,
- Concept II—construction of underground stories with the use of diaphragm walls and prestressing.

Both these technologies will be presented for the same calculation case and the same ground conditions, in order to reflect the cost-estimate function of the described methodology as accurately as possible. Similarly, the unit prices are used here only as an example for the analysis of the described technology. Basic data concerning the discussed example is as follows: building dimensions: 40 × 20 m, the thickness of diaphragm walls: 80 cm, length of diaphragm walls: 20.0 m, story height: 3.0 m, ground conditions: medium sand, no groundwater; the second level of support will be a strutting ceiling with a technological opening (O-ring concept, see Figure 10).

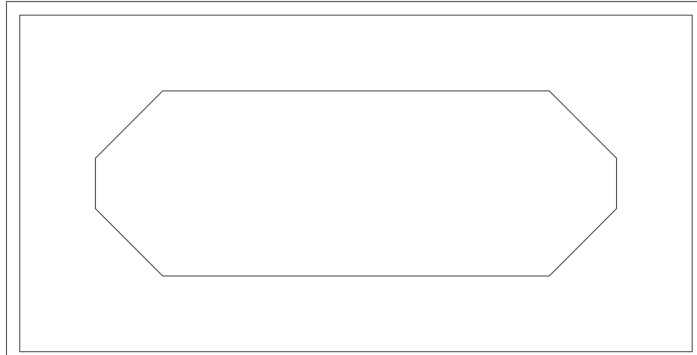

**Figure 10.** Concept Ia, Ib, and II—execution of the second support level with the use of a spreading ceiling with a technological opening—top view. (Own source).

In each of the concepts discussed, the diaphragm wall maintains the same geometry and is made using the same technology. For the three concepts, the reinforcement index was assumed by the authors at the level of 75 kg/m$^2$ of the wall. The current price of reinforcing steel (Q3 2021) is 3.7–3.9 PLN/kg. The area of walls in the discussed example is:

$(40 + 20) \times 2 \times 2 = 2400$ m$^2$. The weight of the reinforcing steel for the discussed example (Concept I): 2400 m$^2$· 75 kg/m$^2$ = 180,000 kg. Assuming the price of 3.8 PLN/kg, the following value was obtained: 3.8 PLN/kg × 180,000 kg = PLN 684,000.

Concept Ia—ground anchors—in this concept, the construction of underground stories is carried out with the use of ground anchors. Performing analysis and calculations allowed for obtaining the system of ground anchors presented in Figure 11.

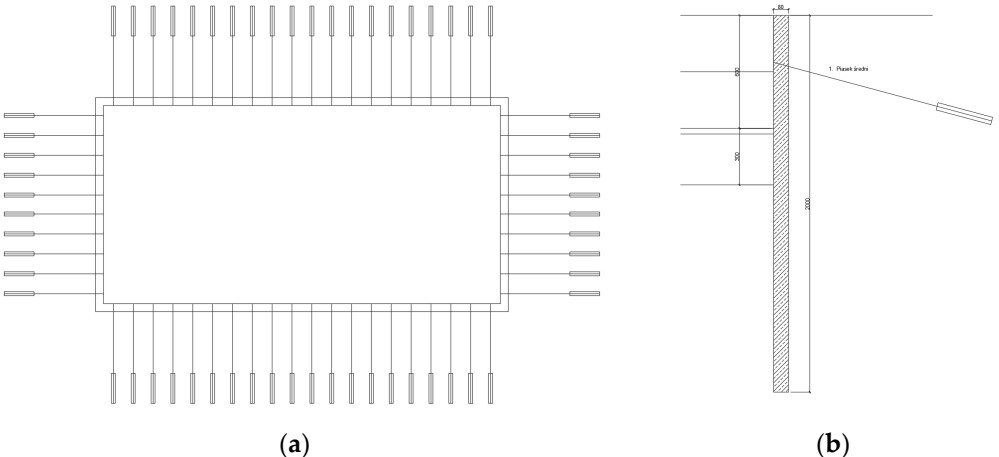

(a)           (b)

**Figure 11.** (**a**) Concept Ia—execution of first-level support with ground anchors, anchor length 18.0 m, distances between anchors, 2.0 m—projection. (**b**) Concept Ia—execution of first-level support with ground anchors—concept cross-section. (Own source).

Concept Ib—steel struts—concept Ib shows a common case of strengthening the walls of a trench with the use of steel struts. In the discussed case, it was proposed to use steel struts with a diameter of 508 × 12.5 mm and 813 × 12.5 mm (Figure 12).

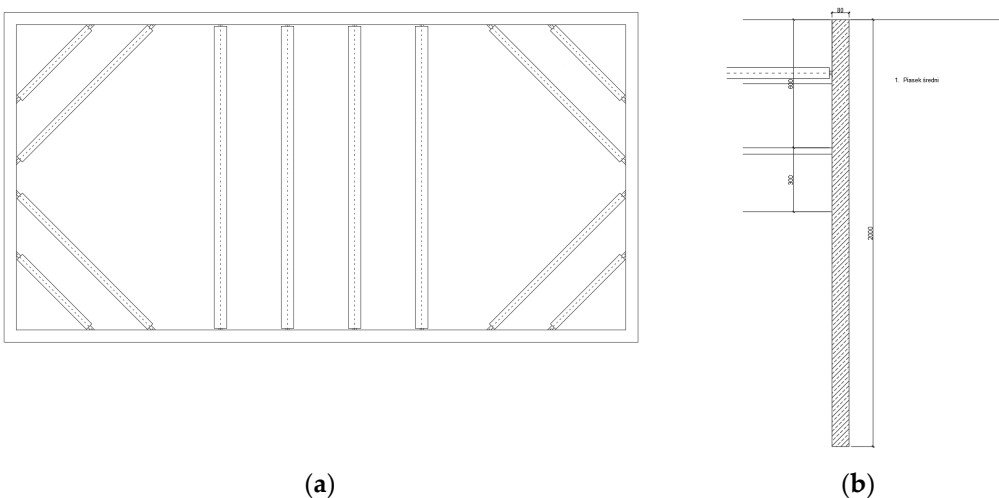

(a)           (b)

**Figure 12.** (**a**) Concept Ib—execution of the first expansion level with the use of steel struts—concept projection. (**b**) Concept Ib—execution of the first expansion level with the use of steel struts—a cross-section of the concept. [own source].

Concept II—innovative approach (see Figure 5)—prestressing of the diaphragm wall.

That innovative concept makes it possible to dispense with the use of ground anchors or steel struts. It all depends on the given conceptual situation, but the construction of walls with no struts is undoubtedly an advantage. It is worth adding that the savings will also appear in the expansion floor itself, because, when using prestressing, the horizontal displacement towards the excavation is reduced, so the normal stresses in the floor will also be lower, which translates into a reduction in the reinforcement quantity used.

It is worth emphasizing that the laying of the prestressing cable can have different effects itself. That feature can be used in many scenarios depending on the given needs, but in this case, the simplest variant was used, i.e., the prestressing cable was laid along the vertical axis of the diaphragm wall. Nonlinear laying of the cable according to the occurrence of the prestressing moments for a given phase, and in general for the envelope translates into a reduction in horizontal displacements and stresses in the reinforced concrete element. A natural consequence is reducing the quantity of steel needed to transfer the tensile stress in the element. Linear laying of the cable or placing it closer to the trench may have a positive effect on the stress system, which is responsible for the fulfillment of the scratch condition in the further design process.

The costs common to all variants, such as making narrow excavations, producing a bentonite suspension, the use of machinery, and the amount of concrete used, are the same. The difference appears when one needs to count the quantity of main reinforcement and steel struts or ground anchors. Additionally, in Concept II, it is possible to make an excavation to a depth of 6.0 m, which is equal to the height of two stories, without interruption for the installation of struts or the execution and prestressing of ground anchors.

In Concept Ia, in the considered case, 60 ground anchors 18.0 m long were used. Costing the ground anchors is very difficult due to the discrepancy in the price of the ground anchor. The prices for the implementation of the ground anchors in 2021 were in the range of 150–320 PLN/m. It all depends on the location of the anchors, soil and water conditions, the number of anchors, the level in relation to the diaphragm wall, whether they will be made under the ceiling or as the first anchoring level, anchor geometry, club length, prestressing force, type of injection, type of anchor, whether the anchors will be permanent or temporary, and the need for additional sealing, etc. Additionally, one should remember the need for repeated mobilization in the case of expansion of the anchors and their injection. Based on the authors' professional experience, it was decided that it would be correct to assume the value of 200 PLN/m (due to the lack of groundwater) for making the anchor as the first anchoring level and making the anchors in non-cohesive soils; 60 anchors × 200 PLN/meter × 18 linear meters = PLN 216,000.

Cost of Concept Ib: In concept Ib, the calculation results showed the necessity of using steel struts with lengths: 6.5 m-4 items-Ø508 × 12.5 mm; 12.0 m-4 items-Ø508 × 12.5 mm; 20.0 m-4 items-Ø 813 × 12.5 mm. The sum of running meters of steel struts: Ø 508 × 12.5 mm: (6.5 m + 12 m) × items = 74 m, strut weight: 153 kg/m-74 m × 153 kg/m = 11,322 kg. The sum of running meters of steel struts-Ø813 × 12.5 mm: 20 m × 4 items = 80 m. Strut weight: 247 kg/m; 80 m × 247 kg/m = 19,760 kg. Total steel weight: 11,322 kg + 19,760 kg = 31,082 kg. The Warsaw market in 2021 dictated the price of 8.0 PLN/kg for the used strut. 31,082 kg × 8 PLN/kg = PLN 248,656.

Cost of the concept II: In order to estimate the costs related to the use of prestressing in the discussed concept, it is necessary to analyze the whole cost of prestressing, including costs of material and additional elements, labor, prestressing, and service, as well as the cost reduction issue related to the lower reinforcement ratio for the diaphragm walls themselves, but also the expansion ceiling. The cost estimate was as follows: the number of prestressing channels was 84 (the calculations assume 2 channels per 2.8 m long section, the number of sections: 42) and the length of the channels was 15 m. Prestressing costs (data for the second quarter of 2021): PLN 150/m—data obtained through consultations with contractors. The total number of running meters of prestressing cables: 84 × 15 m = 1260 m. Therefore, the cost of prestressing was 1260 running meters × 150 PLN/m = PLN 189,000. A very important issue related to the analysis of the use of prestressing is to reduce the mild steel used in the reinforcement cage.

The results of the cost analysis for the discussed examples are presented in Table 2.

**Table 2.** Summary of costs for each of the concepts discussed. [own source].

| Id | Concept | Materials Labor Equipment Summary [PLN] | Use of Third-Party Terrain (for Anchors) [PLN] | Reinforcement in Diaphragm Wall [PLN] | Total [PLN] |
|---|---|---|---|---|---|
| Ia | Ground anchors * | 216,000 | 324,000 | 684,000 | 1,224,000 |
| Ib | Steel struts ** | 186,492 | - | 684,000 | 870,492 |
| II | Prestressing *** | 189,000 | - | 684,000 | 790,920 |

* To the number of ground anchors related to the execution of them (which is the most expensive option anyway), it is necessary to add costs related to the use of the third-party land, when the anchors have to go beyond the construction site. Polish regulations, depending on whether the area is public or private, demand an increase in the cost of execution in this case from PLN 129,600 to even PLN 324,000. For the calculation and comparison of the costs of variants, a scheduled period of 4 months needed for the implementation of the anchors was adopted, assuming that all anchors were made outside the investor's plot (which gives an approximate total amount of PLN 1,126,800 + PLN 324,000). ** When it comes to struts, the re-work of dismantling them should be taken into account with an awareness that it takes the time of the crane and employees. During the assembly of the struts, problems with the connection of the steel strut and the diaphragm wall usually occur. A specially prepared sheet, commonly known as a "steel brand", is mounted in the diaphragm wall for the reinforcement basket. This plate is connected to the bars of a reinforcing basket assembly. The prepared in advance reinforcement cage with the prepared sheet is placed in a ground joint filled with bentonite suspension. Then, diaphragm wall sections are concreted using the contractor method. The sheet must be placed in the right place because after the diaphragm wall is excavated, it will be not possible to correct its position. Therefore, there are problems in the form of adjusting the strut to the previously concreted steel brands. It is only possible to fit these elements together at construction commencement. Because of the risk of mismatching the elements, the contractor protects themself by increasing the costs of installing such struts. *** Depending on the soil conditions, the use of prestressing reduces the displacement of the diaphragm wall crest and influences the envelope of bending moments in the wall. If you want to dimension a wall for a lower bending moment, you will get a lower reinforcement ratio compared to the traditional approach.

The analysis above was carried out with general assumptions in order to reflect as accurately as possible the sense of adapting bridge technologies to retaining structures. It is obvious that each design concept should be considered individually. Particularly when it comes to the geotechnical aspects, it is very important to consider all influencing factors on the price of a given project. The authors' intention was not to analyze a specific example based on the data from a specific contractor but to convey the sense of this innovative solution. Therefore, the authors used only the widely known current prices of materials, and they accepted the labor costs based on consultations with a few contractors when it was necessary for the analysis.

## 6. Conclusions

The article presented an alternative and innovative approach to designing and making diaphragm walls. The Authors considered commonly known methods of deep excavation executions and suggested the use of the prestressing method as a sustainable way to construct underground walls. The comparison shows that the innovative method leads to a reduction in hazards on a construction site with deep excavations, by facilitating avoidance of accidents with people working under steel trusses or making it possible to use cranes when excavations are support-free. Said method is also environmentally friendly because it allows limiting reinforcement steel by the use of prestressing cables.

In the second article in the *Sustainability Journal* of MDPI titled "Modern Methods of Diaphragm Walls Design" the authors present the concept of the calculation methodology for diaphragm walls design.

**Author Contributions:** Conceptualization, M.F. and G.K.; methodology, P.N.; software, M.F.; validation, M.F., G.K., P.N.; formal analysis, G.K.; investigation, M.F.; resources, P.N.; data curation, M.F.; writing—original draft preparation, M.F.; writing—review and editing, P.N.; visualization, P.N.; supervision, G.K.; project administration, P.N. All authors have read and agreed to the published version of the manuscript.

**Funding:** This research received no external funding.

**Conflicts of Interest:** The authors declare no conflict of interest.

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
