# Peer review of "Hazard Reduction in Deep Excavations Execution"

_sustainability, doi:10.3390/su14020868_

Round 1
Reviewer 1 Report
Summary
The paper by Mateusz Frydrych, Grzegorz Kacprzak and Paweł Nowak presents an introduction to a new construction method in which deep excavation walls are prestressed in order to reduce the risks of the excavation without incurring in an increase of used resources. The paper is meant to be a first part preceding a second publication with the calculation method.
Broad comments
- Introduction is poorly written, very general, does not introduce the kind of research to be presented, does not frame the research in the existing literature and contains almost no references.
- When stating the advantages of concrete prestressing, or presenting existing construction methods, references are needed to support the information.
- Conclusions mention: “comparison shows that innovative method leads to hazards limitation…” but no such comparison is present in the manuscript.
- In general, the paper lacks justification of the assumed method advantages with respect to existing methods, the written English needs to be improved, a more thorough bibliographic research needs to be added. As it is announced in the manuscript, a second article is meant to present the calculations of the method. I suggest to merge the two articles in one and pay special attention to the author guidelines and publication criteria.
Specific comments
Line 14: “i.e.” which translates to “that is” is out of place here.
Line 29: What is the soil center?
Lines 59-61: Reference needed
Line 222: however?
Most of minor comments have been omitted because of their abundancy, authors need to revise the entire document.
Author Response
Dear Reviewer, many thanks for your comments, we have corrected the article according to your requirements. We extended introduction, add substantial list of references, explain (with examples and calculations) the advantages (costs for example). We also agreed with Editors that two articles will be not merged. Specific comments and English language improvements - done.
Best Regards,
Paweł, Grzegorz, Mateusz
Reviewer 2 Report
The manuscript is very interesting and will potentially contribute to a new way of thinking when designing a deep excavation. However, the research in question needs to be substantiated with additional facts.
Everyone knows that during a deep excavation in the soil, potentially dangerous stresses are released. I suggest that they be shown and explained what and how much they amount to during this way of deep excavation.
You have explained little about groundwater, which can be potentially dangerous during deep excavation, and how it affects the new method of deep excavation. How can the appearance of groundwater during deep excavation be remedied? Is it meant using submersible pumps or some other method to lower the groundwater level?
You also mention the important role of geodesy in the process of deep excavation. You did not mention what geodetic methods are used to monitor such a deep excavation. Is it perhaps by the installation of reference benchmarks, how is the displacement, clinometry controlled, and how is the height and subsidence of the subject diaphragm controlled, perhaps by leveling measurements.
Overall, the work is interesting but needs further refinement to substantiate all the methods described.
Author Response
Dear Reviewer, many thanks for positive comments. We have addressed your detailed comments by addition of issues related to ground water problem in deep excavations, as well as paragraph on geodesy. Correctness of English langauge was also significantly improved.
Best Regards,
Paweł, Grzegorz, Mateusz
Reviewer 3 Report
The idea of vertical prestressing of reinforced concrete walls is derived from the implementation of, for example, prestressed columns, which are of increasing length and slenderness. Such a solution requires high accuracy in the execution of the supporting zone, good adhesion of the cable - tendon to concrete, effective anchoring of the tendon on the active side. Adjusting the track for the tendon position in the case of high walls is a difficult task for walls concreting into the ground. In addition, there are calculations of the losses in prestress of prestressed concrete. Corrosion protection when concrete pouring is out of control in diaphragm walls several storeys high.
The authors refer to another paper, where they continue the description of the proposed method, but by comparing different techniques of construction of diaphragm walls, they should describe the methods of solving the above-mentioned issues. The construction of diaphragm walls using the prestressing technique increases the difficulty of implementation to the same extent as the transition from reinforced concrete to prestressed structures. The cost of execution is similar.
Authors should add some informations about above shown problems.
Author Response
Dear Reviewer, many thanks for your comments and help in paper improvements. Our further research will be related to tendon position in the wall, although we were trying to explain the problem in the second article, which - as agreed with editor - is mentioned in the article text (in the abstract - and not as bibliography to avoid self citation). We provided also information about the advantages of the method, taking under consideration technological issues (like easy crane operation, when no struts used) and costs. Correctness of English language was also significantly improved.
Best Regards,
Paweł, Grzegorz, Mateusz
Reviewer 4 Report
Dear, Authors
1. The abstract should be containing some results.
2. English language and style are minor spell check required.
3. The authors should specify the objectives of the test matrix.
Author Response
Thank you very much for acceptance and remarks. We intruduced the adjustments.
1. The abstract should be containing some results – abstract corrected.
- English language and style are minor spell check required. We did our best to improve English language in our paper.
- The authors should specify the objectives of the test matrix. Objectives specified.
Round 2
Reviewer 1 Report
Review report, round 2:
- Language used in the introduction is still not clear: for example (but not limited to), in lines 26-28: “e.g. in the case of diaphragm walls are part of the slab-pile foundation, so it is impossible to miss this important process of their design and implementation on a construction site.” It is not clear if this refers to the particular case of diaphragm walls being part of the slab-pile foundation, or the particular case is the diaphragm walls themselves. The end of the sentence is not clear either, what “impossible to miss” means? Does it mean that it should not be overlooked, or that it is so obvious that no one would ever miss it?
1b, lines 77-79: Why the effect of pressure can be excluded due to consolidation phenomena? What is the relation of soil consolidation with the pressures exerted on the wall?
- Section 2, first paragraph: line 101 “this technology”, line 103 “this method”, no technology or method have been introduced, so the reader does know what those statements are referring to. It seems that the ideas of ground identification and concrete prestressing are mixed in the paragraph.
- New paragraph in lines 374-378 talks about “both models” but no models have been previously introduced.
- Numerical model results are presented in section 4 before the model description which is done in section 5.
- There is no description of the used numerical model at all, assumptions, boundary conditions, constitutive equations, solving method, material properties (including soil and structural elements). A table comparing Plaxis and Sofistik results is provided as a validation, it is hard to judge whether this validation is correct without knowing what are the developments done by the authors in Sofistik, or what is the problem being solved.
- Overall, the English is still poor and this makes it difficult to follow and understand the paper. The order of the presentation is not logic, models are mentioned before being introduced, numerical results are presented before the problem is described. The positive point is the addition of section 5, which was lacking in the first revision round, and is the output in which this research should be focused.
Specific comments:
Line 74: “can continued.” Not correct
Line 102: “which” added is not correct
Table 1. m and mm labels seem to be wrong, or both models give really different results
Author Response
Dear Sir or Madam,
Many thanks for your second revision.
RE.1 and 6: International English will never satisfy native speaker. Few other revievers, including academic one – accepted the language as sufficient for international reader. Anyway, we checked our English language again and did necessary improvements (grammar, spelling, punctuation, etc). Thank you for your comments.
RE. 4: In chapter 4 results were presented from numerical ascpects, because it was necessary to check idea of new suggested diaphragm prestressing method and in chapter 5 model description was presented and it was possible to use this method in practice case.
RE. 5: Description of the used numerical model assumptions (boundary conditions, consti-tutive equations, solving method, material properties (including soil and structural ele-ments) are given in details in the article “Modern Methods of Diaphragm Walls Design” Sustainability. 2021. Vol. 13, no. 24, p. 1–14. DOI 10.3390/su132414004. (presented by the same authors).
RE. 6: Many thanks for your possitive comments about the addition of section 5.
Specific comments: Many thanks. All corrected.
Best Regards
Gregor, Mateusz and Paul
Reviewer 2 Report
Accept in present form.
Author Response
Many thanks for the acceptance. We did our best to improve English language in our paper.